biotechnology/microbiology/environmental science

azo dye, metabolite by-product, biodegradation, biotransformation, xenobiotic

**Author for correspondence:**
Khanom Simarani
e-mail: hanom_ss@um.edu.my; hanomks@gmail.com

This article has been edited by the Royal Society of Chemistry, including the commissioning, peer review process and editorial aspects up to the point of acceptance.

# Comparative static and shaking culture of metabolite derived from methyl red degradation by *Lysinibacillus fusiformis* strain W1B6

## Ira Puspita Sari and Khanom Simarani

Institute of Biological Sciences, Faculty of Science, University of Malaya, 50603 Kuala Lumpur, Malaysia

IPS, 0000-0002-6886-3481; KS, 0000-0002-0270-1324

This paper reports on the comparative characteristics and properties of the metabolites derived from methyl red (MR) decolorization by *Lysinibacillus fusiformis* strain W1B6 under static and shaking conditions. A batch culture system was used to investigate the effect of aeration on azoreductase activity in the biodegradation process, transformation of colour removal and the metabolite products. Biodegradation analysis was monitored using Fourier transform infrared spectroscopy and high-performance liquid chromatography while metabolites were determined using gas chromatography–mass spectroscopy. Phytotoxicity and anti-microbial tests were also conducted to detect the toxicity of metabolites. The results showed that this strain grew more rapidly under shaking conditions while azoreductase activity increased more rapidly under static conditions. Despite that, no significant difference in the decolorization was observed under both static and shaking conditions with up to 96% and 93.6% decolorization achieved, respectively, within 4 h of incubation. MR was degraded into two fragmented compounds, i.e. 2-aminobenzoic acid and *N,N*-dimethyl-1.4-benzenediamine. The concentration of 2-amino benzoic acid was higher under static conditions resulting the biotransformation of 2-amino benzoic acid into methyl anthranilate more rapidly under static conditions. Other metabolites were also detected as intermediate biotransformation products and by-products. Less or no toxic effect was found in the metabolite degradation products under both culture conditions.

# 1. Introduction

Azo dye degradation by bacteria mostly occurs in two stages: firstly, reductive cleavage of the azo linkages resulting in the potentially hazardous aromatic amines—usually under anaerobic conditions; and secondly, degradation of the aromatic amines—which is almost exclusively an aerobic process [1]. The mechanism of the redox-mediator-dependent reduction of azo dyes using whole bacterial cells under anaerobic conditions has been described previously. In aerobic environments, the reduction of azo dye is inhibited by the presence of oxygen because the reduced redox mediator has a higher preference for oxygen than the azo dye [2].

Azoreductase has a primary role in cleaving the azo bonds (-N=N-) to produce colourless aromatic amine in methyl red (MR) degradation [3]. It is one of the most promising enzymes in azo dye remediation [2] and acts as a highly efficient biocatalyst in MR biodegradation [4]. Azo bonds in MR are broken down into two main metabolic by-products: $N,N^1$ dimethyl-$p$-phenylenediamine (DMPD) and 2-aminobenzoic acid (2-ABA) [3,5]. A previous study revealed that DMPD formation remains integrated into the culture and is mutagenic in nature [5]. The end products of azo dye degradation are influenced by the type of microorganisms [6] and physico-chemical parameters applied during the cultivation [7,8]. Previous studies reported that static conditions are preferred and are more efficient for azo dye decolorization [9,10], and the aromatic amines of its metabolite products were assumed to resist further degradation compared with shaking cultivation conditions [11,12]. However, the effects of different modes of cultivation (i.e. static and shaking conditions) on dye decolorization efficiency, azoreductase activity and metabolite products of biodegradation still need to be investigated.

The aim of this study was to investigate the metabolite produced by *Lysinibacillus fusiformis* strain W1B6 under shaking and static cultivation conditions such as indicated in figure 1. The decolorization efficiency and enzymatic activities that play a role in degradation were also determined. This study could contribute to the understanding of metabolomics characterization of microbial bioremediation as an important aspect of environmental biotechnology.

# 2. Material and methods

## 2.1. Dyes and chemicals

MR was purchased from BDH Chemical Ltd. (UK) and used without any further purification. Reduced nicotinamide adenine dinucleotide (NADH), DMPD and 2-ABA were purchased from Sigma-Aldrich (USA). All other chemicals were of analytical grade and highest purity.

## 2.2. Microorganism and culture condition

The microorganism used in this study was isolated from raw textile wastewater and identified as *L. fusiformis* strain W1B6 (accession no: KP233829) [13]. The strain was maintained and sub-cultured monthly on fresh nutrient agar (NA) slants and stored at 4°C. The culture was also kept in sterile glycerol stock (40% v/v) at −20°C. Nutrient broth (NB) was used to reactivate the cell growth for inoculum preparation.

## 2.3. Experimental design

Biodecolorization was carried out in 250 ml conical flasks containing 100 ml of mineral salt medium (MSM) with 100 mg l$^{-1}$ of MR and 0.1% (w/v) of yeast extract. The flasks were incubated at $30 \pm 2$°C for 4 h [13]. The experiments were conducted under static and shaking conditions at an agitation rate of 120 r.p.m. Samples were collected at intervals of 1 h until the equilibrium stage was reached. Samples were then centrifuged at 10 000 r.p.m. for 20 min. Cell mass and supernatant were used for further analysis. Control was conducted with no bacteria culture.

## 2.4. Biodegradation analysis

The biodegradation spectrum of cell-free supernatant was monitored using a Jusco v-630 spectrophotometer at 350–700 nm. The percentage of dye decolorization was determined using the following equation [14]:

$$(\%) \ \text{decolorization} = \frac{\text{OD}_0 - \text{OD}_t}{\text{OD}_0} \times 100 \, ,$$

(2.1)

**Figure 1.** Proposed scheme of MR biotransformation by *Lysinibacillus fusiformis* strain W1B6.

where $OD_0$ is the initial dye concentration and $OD_t$ is the dye concentration after a predetermined incubation time ($t$).

The potential azoreductase enzyme secreted as a result of microbial action involved in the decolorization was determined spectrophotometrically at 437 nm, according to Sari & Simarani [13]. Enzyme activity was calculated with an extinction coefficient of 23.36 mM$^{-1}$ cm$^{-1}$.

## 2.5. Determination of metabolites

Cell-free supernatant was extracted using an equal volume of ethyl acetate, dried over anhydrous sodium sulfate ($Na_2SO_4$) and evaporated using a rotary evaporator [13]. The dried samples were dissolved in high-performance liquid chromatography (HPLC)-grade methanol for further analysis. The metabolites generated were further analysed using Fourier transform infrared spectroscopy (FTIR) and HPLC. The metabolite by-products were then identified using gas chromatography–mass spectroscopy (GC-MS) spectra.

The FTIR analysis was conducted using a Spectrum 400 (Perkin Elmer) spectrometer in the IR region of 450–4000 cm$^{-1}$ with 16 scan speed. The dried sample was prepared for pellet formation using spectroscopic pure KBr and fixed for analysis.

The HPLC analysis was carried out in an isocratic Waters™ 486 Tunable Absorbance Detector using a $C_{18}$ column of 100 mm length × 4.6 mm and 5 µm particle size. The mobile phase was acetonitrile:acetate buffer (50 mM, pH 3) = 50 : 50 (v/v) with a flow rate of 1 ml min$^{-1}$, injection volume of 10 µl, at room temperature. The detection was monitored at a wavelength of 254 nm [4].

Further identification of metabolites was performed using a QP2010 gas chromatograph coupled with a mass spectrometer [15]. The ionization voltage was 70 eV. GC was conducted in the temperature programming mode with an RTX-5 column (Restek Corporation, Bellefonte, PA, USA) (0.25 mm, 30 m, 0.25 µm). The initial column temperature was pre-set as follows: 80°C for 2 min, then increased linearly at 10°C min$^{-1}$ to 280°C, and held for 7 min. The temperature of the injection port was 280°C, and the GC–MS interface was maintained at 290°C. Helium was used at a flow rate of 1.0 ml min$^{-1}$ as the carrier gas. Degradation products were identified by mass spectra, retention time (RT) and fragmentation pattern in the National Institute of Standards and Technology (NIST) spectral library stored in the computer software (v. 1.10 beta, Shimadzu) of the GC–MS. The spectra were recorded with a scanning range of 45–500 $m/z$ and minimum abundance of 10 000. The identified compounds were recorded according to per cent area, and the compound with the highest area was designated as the main product. Culture with no MR as the negative control was used in the metabolite analysis.

## 2.6. Toxicity test studies

Two types of toxicity test were performed in order to evaluate the toxic impact of the metabolite extract of MR degraded by *L. fusiformis* strain W1B6. The extracted metabolites of MR degradation from static and shaking culture were dried and dissolved in sterile distilled water to form a final concentration of

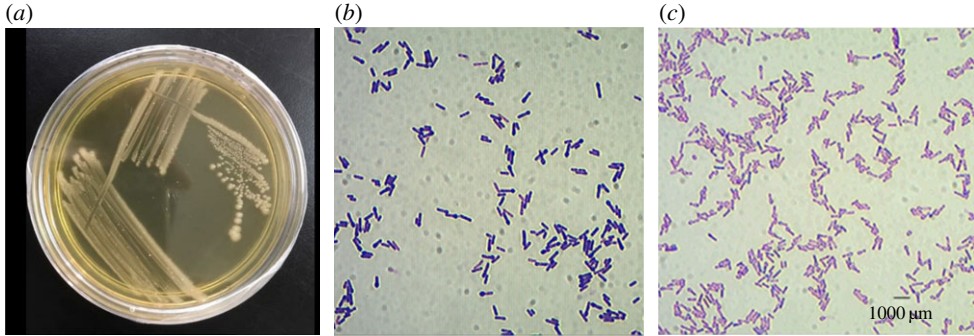

**Figure 2.** Photomicrograph of cell morphology. Cell morphology of *Lysinibacillus fusiformis* strain W1B6 on NA (*a*), Gram reaction (*b*) and spore staining (*c*) under light microscope (1000× magnification).

100 ppm. The control MR was used as a comparison. Toxicity studies of the control MR and its degradation product were also carried out.

The phytotoxicity test was done under ambient laboratory conditions using common agricultural crops: *Vigna radiata* (mung bean), *Vigna unguiculata* (black-eyed pea) and *Arachis hypogaea* (peanut). Ten seeds of each crop were cultivated in a Petri dish and watered daily using 7 ml of control MR and its degradation product (100 ppm). Sterile distilled water was used as a negative control. Each experiment was conducted in triplicate. The germination rate, plumule length and radicle length of the seedling were observed after 7 days.

The microbial toxicity test was carried out using an agar diffusion test [16] to evaluate the toxicity of the metabolites extracted from MR degradation. Tetracycline (50 µg l$^{-1}$) and an aqueous solution of 0.9% (w/v) NaCl were used as positive and negative controls, respectively. The target microorganisms, i.e. *Staphylococcus aureus*, *Bacillus megaterium*, *Escherichia coli* and *Pseudomonas aeruginosa*, were grown in NB for 18 h to reach an OD of 0.5 on the McFarland scale. A volume of 100 µl of the bacterial suspension was uniformly spread on the surface of an NA plate using a sterile cotton swab. Wells of 8 mm diameter were punched into the medium and 50 µl of each test solution was added into the respective wells under sterile conditions. Each solution was tested in triplicate and the inhibition zones were measured after 24 h incubation at 30 ± 2°C.

## 2.7. Statistical analysis

All experiments were performed at least in triplicate. The numerical results are expressed as the mean ± s.d. Data were analysed by one-way analysis of variance (ANOVA) with the Tukey–Kramer multiple comparisons test using IBM SPSS Statistics 23. Readings were considered significant when $p$ was less than or equal to 0.05. Statistical analyses were also performed using Sigma Stat 12.0.

# 3. Results and discussion

## 3.1. Microorganism profile

The morphological characteristics of *L. fusiformis* strain W1B6 on the NA plate were observed based on the colony shape such as form, elevation, margin and size. The colony appearance of its surface, texture, colour and opacity were shiny and smooth, dry, creamy and opaque, respectively (figure 2*a*). Gram staining showed a Gram-positive reaction and rod shape with an oval terminal endospore, resembling a tennis racket (figure 2*b*). Some mature endospores were released from the vegetative cell as free endospores (figure 2*c*).

## 3.2. Decolorization study under different batch culture

Maximum dye decolorization under both static (96.25%) and shaking (93.62%) culture conditions of *L. fusiformis* strain W1B6 was achieved after 4 h of incubation. Figure 3*a* shows the time profiles of coloration and cell mass. The result revealed that cell mass under shaking conditions was higher than under static conditions due to the mixing, which provided more O$_2$ transfer coefficient that may be

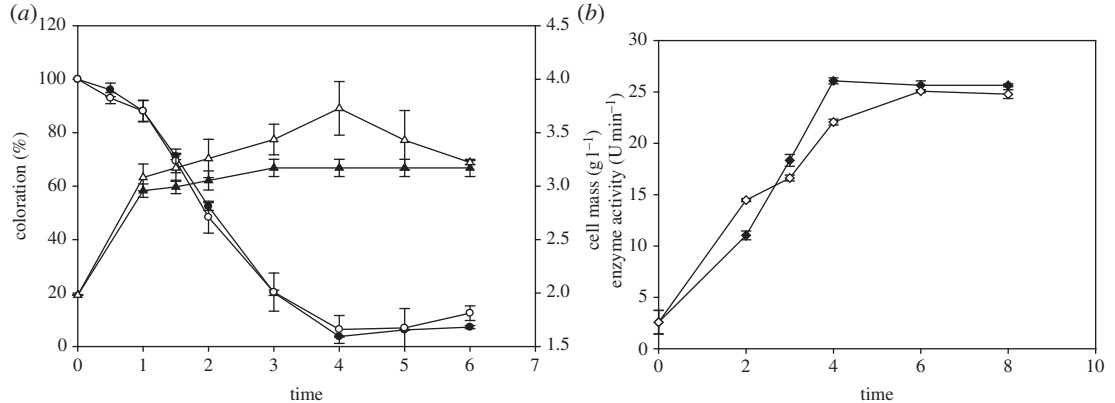

**Figure 3.** Decolorization profile. Coloration and cell mass (*a*) and azoreductase activity (*b*) by *Lysinibacillus fusiformis* strain W1B6 by time, under both static and shaking conditions. Symbols: (filled) static, (open) shaking, (triangle) cell mass, (circle) coloration and (diamond) enzyme activity.

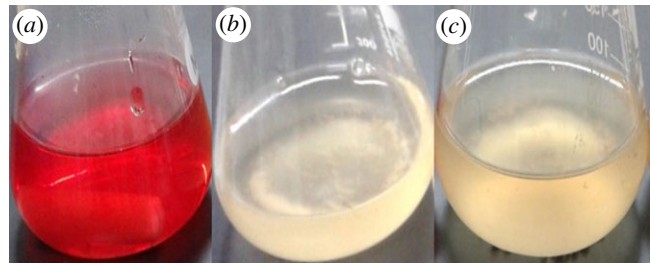

**Figure 4.** Dye decolorization after 4 h incubation. Control MR (*a*) and using *Lysinibacillus fusiformis* strain W1B6 under static (*b*) and shaking (*c*) culture conditions.

required for microbial growth. The efficiency of decolorization under the static conditions became slightly higher after 4 h of incubation. Although a lower growth rate was observed under static culture, the decolorization was complete under this condition. However, under shaking conditions, azo dyes are normally resistant to attack by bacteria owing to the inhibition of azoreductase by the presence of oxygen [17]. However, no significant differences in decolorization efficiency were observed under both shaking and static conditions, probably because of the presence of enzymes that catalyse the degradation of MR.

The profile of azoreductase activity, which plays an important role in the decolorization process, is shown in figure 3*b*. Both decolorization and azoreductase activity followed a similar trend in the early stages under the two modes of cultivation. The azoreductase activity increased rapidly and reached a peak after 4 h under the static condition. The activity was however slightly slower under the shaking condition where the peak was attained only after 6 h. The rapid degradation of MR by the bacteria seen in the initial stages could be due to the induction of higher azoreductase secretion in the cell by the dye. This is supported by a report that bacterial cells produced azoreductase either as a constitutive enzyme under control conditions without dye or as an inducible enzyme with dye treatment [13].

During the first 2 h, higher enzyme activity in the shaking condition was associated with rapid cell growth. This result is similar to a previous report that increased azoreductase activity was associated with increased cell density of *Bacteroides fragilis* [18]. However, a continuous shaking regime could provide excess oxygen, which may inhibit the azoreductase activity and decrease the dye reduction where oxygen could provide a more positive redox potential. This finding corroborates the study on *Aeromonas jandaei* in MR decolorization where bacterial cell growth and the decolorization rate around 2 h were slightly higher under aerobic conditions but then dye decolorization was more effective under anaerobic conditions [19]. Similarly, Bragger *et al.* [18] reported that the inhibition of azoreductase activity occurred when *B. fragilis* was in a bubbled system with air rather than nitrogen.

The qualitative degradation of colour in the flask system is shown in figure 4. Static flask culture exhibited less coloration than the shaking flask culture. In order to clarify the decolorization quantitatively and verify the degradation, UV–visible spectroscopy, HPLC and FTIR analysis of both cultures was conducted.

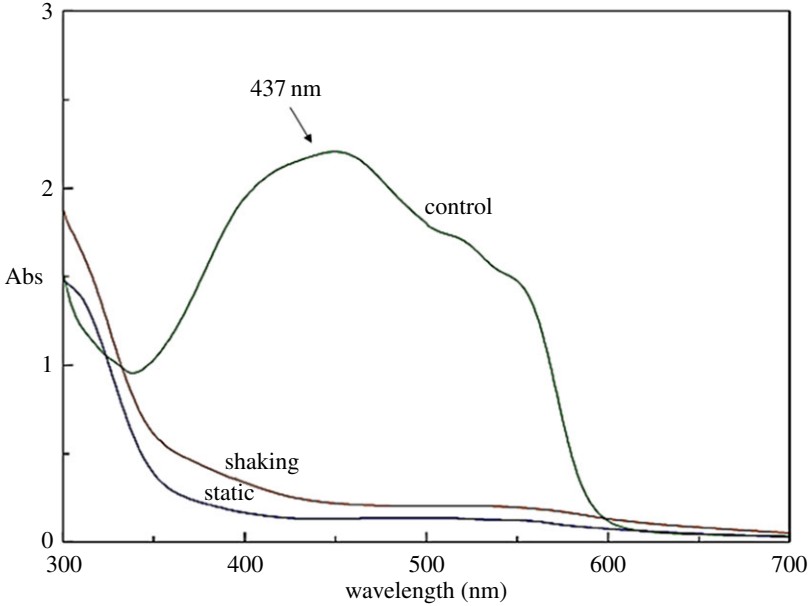

**Figure 5.** UV–vis spectra. MR biodecolorization by *Lysinibacillus fusiformis* strain W1B6 under static and shaking conditions.

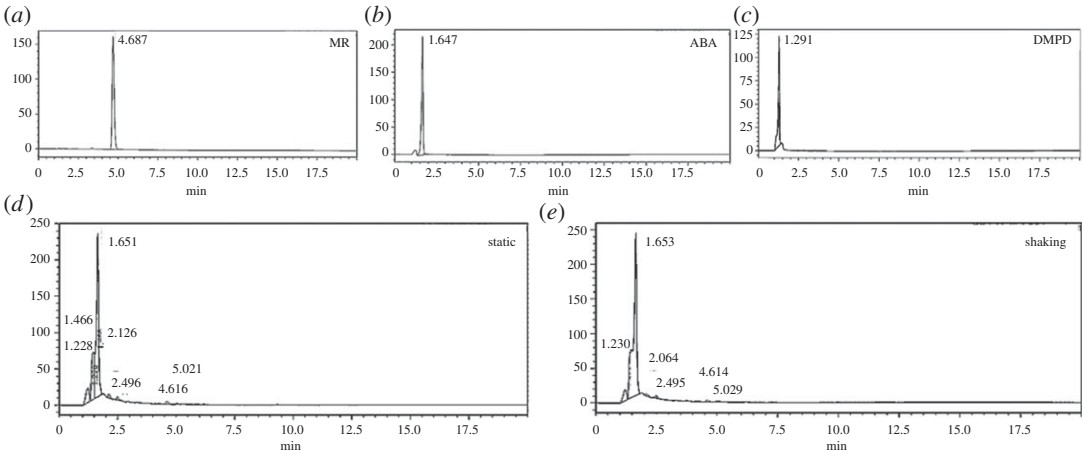

**Figure 6.** HPLC chromatogram. Chromatogram of HPLC of: MR (*a*); standard ABA (*b*) and standard DMPD (*c*); extracted MR degraded metabolite products under static (*d*) and shaking (*e*) conditions after 4 h contact time.

## 3.3. Biodegradation analysis

The biodegradation spectrum of cell-free supernatant is presented in figure 5. The spectra (300–700 nm) of the cell-free supernatant showed that decolorization occurred in line with the decrease in MR concentration under both static and shaking conditions after 4 h. As pointed out by Sari & Simarani [13], MR biodegradation was indicated by either the disappearance of the major absorbance peak or the presence of a new peak.

The cleavage of the azo bond and formation of metabolites from the degraded MR can be confirmed with HPLC, FTIR and GC-MS. The HPLC profiles of MR degradation components under both static and shaking conditions are shown in figure 6. The single peak at an RT of 4.69 min in the control solution corresponds to MR. The two chemical compounds which were used as standard markers were 2-ABA and DMPD with RTs of 1.65 and 1.29 min, respectively (figure 6*a*–*c*)

The peak of MR dye almost completely disappeared after 4 h in both static and shaking bacterial treated cultures (figure 6*d*,*e*). The degradation of MR based on the peak area calculation was 98.82% and 98.72% for static and shaking conditions, respectively. The new compounds released after biodegradation were indicated by the presence of new peaks detected at RT 1.23, 1.47, 1.65, 2.13, 2.50, 4.62 and 5.02 min under static culture and 1.23. 1.65, 2.06, 2.50, 4.61 and 5.03 min under shaking

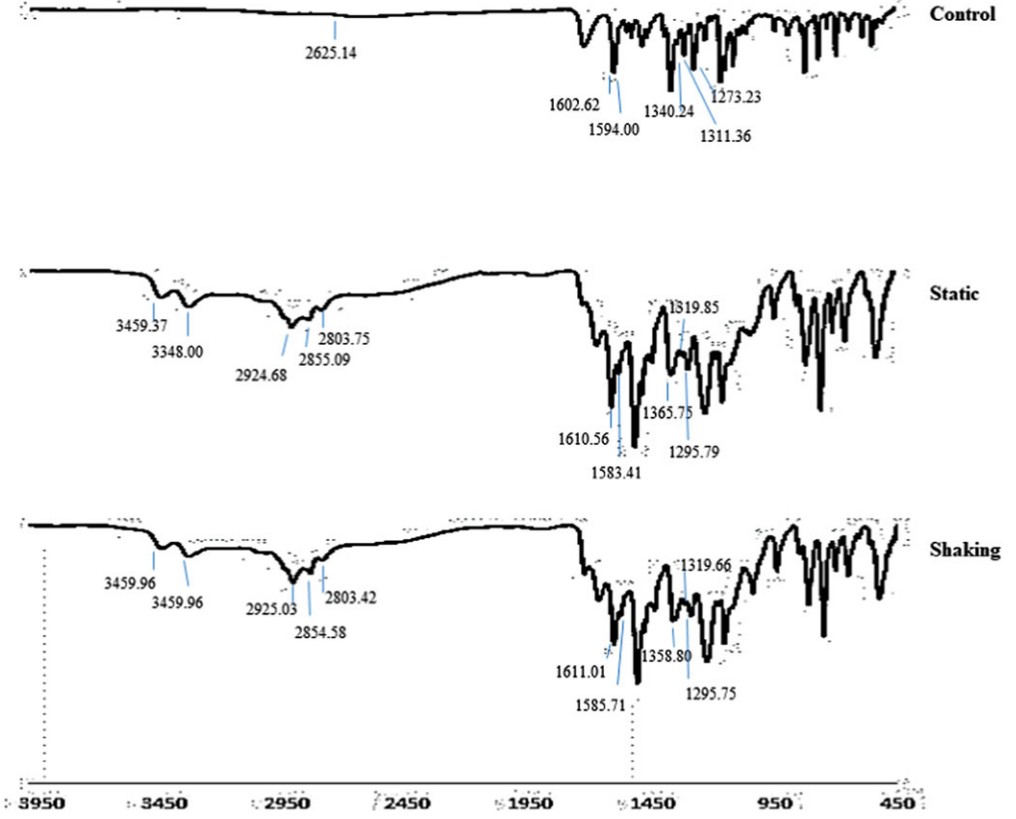

**Figure 7.** FTIR spectra of MR before (control) and after degradation by *Lysinibacillus fusiformis* strain W1B6 under different culture conditions (static and shaking).

culture. The RT of the main peak in both treatment conditions was 1.65 min and this corresponded with the ABA standard (1.65 min), while the RT of another peak (1.23 min) corresponded to the DMPD standard (1.29) in both treatments. The peaks of RT 4.62 and 4.61 under static and shaking condition, respectively, corresponded with MR (4.69). The different compounds/fragments generated after a biodegradation process depend on the type of microorganisms and physico-chemical conditions applied. It was reported that no other peak was observed except MR, 2-ABA and DMPD from degradation of MR by *Klebsiella* spp. and *Acetobacter* spp. [5]. Another report revealed 2-ABA and DMPD as the main metabolic by-products released from MR degradation by *Bacillus* spp. [4]. By contrast, other metabolic compounds were found during the degradation of MR by *Galactomyces* spp. at different temperatures [7].

The functional group analysis was reported by using FTIR spectra of the extracted metabolites under static and shaking conditions, as shown in figure 7. New absorption bands under static (3459.37 and 3348.00 cm$^{-1}$) and shaking (3459.96 and 3349.30 cm$^{-1}$) culture were observed. These bands indicated the stretching of primary $NH_2$ from the reductive cleavage of the azo bond (-N=N-), which might result from the amine group. The assignment of group frequency of 3570–3200 cm$^{-1}$ represents the hydroxyl group, H-bonds and OH stretch. While the appearance of 3510–3460 cm$^{-1}$ and 3415–3380 cm$^{-1}$ under static and shaking conditions were consistent with aromatic primary amine and NH stretch, respectively. The shift of the FTIR spectra of extracted metabolites after 4 h cultivation under static and shaking conditions indicated the biodegradation of the dye into fragmented compounds. According to Coates [20], some functional groups showed a shift in their peak bands, i.e. primary amine-NH at 1650–1590 cm$^{-1}$, aromatic primary amine-CN stretch at 1340–1250 cm$^{-1}$, aromatic tertiary amine-CN stretch at 1360–1310 cm$^{-1}$ and open-chain azo (-N=N-) at 1630–1575 cm$^{-1}$ (figure 7).

## 3.4. Metabolite properties

Extracted metabolite by-products in the strain culture without MR were used as the control to differentiate between the metabolites under MR treatment. Table 1 shows the extracted metabolite compounds of MR degradation under both static and shaking culture conditions, according to the

**Table 1.** Metabolite degradation of MR from GCMS result confirmed by NIST library.

| | static | | | | shaking | | |
|---|---|---|---|---|---|---|---|
| | retention time (min) | m/z | name | | retention time (min) | m/z | name |
| I | 9.998 | 137 | 2-aminobenzoic acid (anthranilic acid) | | 10.052 | 137 | 2-aminobenzoic acid (anthranilic acid) |
| II | 9.238 | 151 | 2-aminobenzoic acid, methyl ester (methyl anthranilate) | | 9.477 | 136 | N,N-dimethyl-1,4-benzenediamine |
| III | 9.470 | 136 | N,N-dimethyl-1,4-benzenediamine | | 13.271 | 268 | phenol, 2-(5-(2-methoxyphenyl)-(1,3,4) oxadiazol-2-yl) |
| IV | 13.265 | 193 | propenal, 3-hydroxy-2-(2-nitrophenyl)- | | 14.626 | 195 | 2-amino-4-methoxymethyl-6-methyl-3-pyridinecarboxamide |
| V | | | | | 9.240 | 151 | 2-aminobenzoic acid methyl ester (methyl anthranilate) |

NIST library in GC-MS recording. The results revealed some major peaks, the masses of which were lower than those of MR (269 *m/z*). The metabolite compounds represented the main products of parent dye fragmentation, intermediate biotransformation and metabolite by-products.

The main extracted fragmented product of the MR degradation with an *m/z* value of 137 was identified as 2-ABA. Another product of fragmentation was DMPD with mass 136 (*m/z*).

According to a standard curve of the compounds, 100 ppm of MR was degraded into 2-ABA and DMPD up to 39.5 and 4.2 ppm under static culture and 24.1 and 7.1 ppm under shaking culture, respectively. The concentration of 2-ABA compounds was higher under both conditions. As shown in table 1, 2-ABA is considered a main product of degraded MR in both conditions, followed by DMPD in shaking culture. However, DMPD was listed as the third compound under the static condition. These results were also consistent with HPLC analysis, which showed 2-ABA as the main peak and DMPD as a secondary product of degradation of MR. It was also reported that aromatic amines are the main biotransformation products of azo dyes [21] because of the reductive metabolism step involved in the cleavage of the azo linkage, attended by the absence of dye colour.

Other extracted compounds such as methyl anthranilate (2-aminobenzoic acid, methyl ester) as an aromatic amine, Propenal, 3-hydroxy-2-(2-nitrophenyl)-), 2-amino-4-methoxymethyl-6-methyl-3-pyridinecarboxamide and Phenol, 2-(5-(2-methoxyphenyl)-(1,3,4) oxadiazol-2-yl) with mass 151, 193, 195 and 268 (*m/z*), respectively, were also released after 4 h of incubation time (table 1). They might be considered as intermediate biodegradation compounds of MR. Methyl anthranilate was the second main metabolite of degraded MR under static culture. This indicated that 2-ABA had already transformed to this compound more rapidly under static culture than under shaking culture because of the higher concentration of 2-ABA under static than under shaking conditions. Methyl anthranilate is one of the volatile carboxyl methyl esters that are synthesized as a secondary metabolite when bacterial expressed proteins with methyl-tranferase activity with high specificity for 2-ABA are exposed to it as a substrate. The regulation of its production probably depends on enzyme concentration and the availability of the substrate [22].

Besides the fragmentation product of MR and its biotransformation, other metabolites were also identified by GC–MS (data not shown). The organic compounds classified as saturated methyl and ethyl esters of fatty acids were abundant at an RT of 14–25 (min). They were the main metabolites in both the control (without MR) and MR treatment. Other metabolites with an RT from 5 to 23 (min) were classified by structural criteria as: *n*-alkanes, 1-alkanols, wax esters and sesquiterpenes.

These studies showed the effect of aeration on the capabilities of the bacterial strain in the decolorization and degradation of MR dye, azoreductase enzyme activity and characterization of metabolites. The finding of this study showed no significant difference in percentage of dye decolorization with different concentrations of the main products and fragmentation compounds under static and shaking culture conditions. There is no report on MR degradation in both aerobic and anerobic cultures by an organism excepting the report on suspended cells of *A. jandaei*, which found the same decolorization rate with DMPD as the only main product of MR degradation under both conditions [19]. However, some reports recorded that anaerobic conditions give a higher efficiency in dye decolorization with the hazardous metabolite product, while aerobic conditions inhibit the reduction of dye [2,18,23,24].

The aromatic amines were transformed by oxidative and reductive metabolism. Reductive cleavage of the azo bond was probably the main toxicological metabolic reaction of azo compounds [8]. The main MR fragmentation and its by-product produced may be more or less toxic than the parent compounds created with the capability of azoreduction on the toxic or carcinogenic level of the dyes [25]. A further study was conducted to evaluate the toxicity of parent dye and its biotransformation metabolites. Changing chemical structures might cause the formation of new xenobiotic compounds which are more or less toxic to the environment.

## 3.5. Metabolite biotransformation pathways

Beside the reductive metabolism, there are three kinds of oxidation pathways of azo dye degradation, i.e. (1) C-hydroxylation following change to a phenol; (2) N-hydroxylation at the primary or secondary amino groups followed by esterification where the activated esters are water soluble, able to be excreted or split off with the –NH⁺ formation and covalently bind to a nucleophilic group of the DNA, and (3) demethylation with the stepwise oxidation of the methyl groups of the dialkylamino compound and the *N*-hydroxy derivative. The methyl groups of dialkylamino compounds are further methylated or react to form a nitrenium compound [26].

The formation of DMPD in this study was probably metabolized by *N*-demethylation where the per cent of the peak area under shaking culture was higher than that under static culture. Phenol, 2-(5-(2-

**Figure 8.** Prediction pathway of MR biodegradation by *Lysinibacillus fusiformis* strain W1B6.

methoxyphenyl) -(1,3,4) oxadiazol-2-yl) was found to be the third major peak area in shaking culture probably due to the oxidation metabolism of C-hydroxylation following the change to a phenol.

These suggest that a reductive mechanism is the main pathway of biodegradation in both static and shaking conditions while the oxidation pathway significantly occurred in the shaking condition. The capability of *L. fusiformis* strain W1B6 to degrade azo dye has thus been confirmed in both conditions. This bacterium secretes oxidoreductive enzyme to reduce the toxic and mutagenic by-products in the biotransformation reaction. The biotransformation reaction may occur for as long as the bacterium secretes the oxidoreductive enzymes. Biotransformations are an important aspect of the fate of xenobiotics, which is often referred to as detoxification [27]. Their metabolic products may be more toxic or less toxic than the parent compounds [25].

Lately, the possibility of azo dyes breaking down and generating amines as a health hazard has been considered by the scientific community. While aromatic amines were the main biotransformation products of azo dyes, they are transformed by oxidative and reductive metabolism. Reductive cleavage of the azo bond was probably the main toxicological metabolite of azo compounds [28].

The prediction pathway of MR biodegradation is shown in figure 8.

Undoubtedly, the ability of bacteria to reduce the dye and the aromatic metabolites to toxic intermediates is a key part of the potential effectiveness in azo dye biodegradation. The final elimination of these xenobiotics from the environment could be carried out through either biotic or abiotic means [29].

Basically, all organisms respond unavoidably to foreign chemicals, called xenobiotics, which include both synthetic and natural products such as pollutants, industrial chemicals, pesticides, secondary metabolites and toxins produced by any organisms. Therefore, the elimination of xenobiotics often depends on their conversion from lipophilicity (easier to absorb) to hydrophilicity (easier to excrete) by enzymes that act as the biotransformation catalyst [30]. This theory agrees with our results. The bacteria produced some oxidoreductive enzymes in transforming the toxic dye into favouring physical properties which are suitable to accelerate the elimination process. Some chemicals stimulate enzyme production in xenobiotic biotransformation. The elimination biotransformation process occurred continuously in the flask  system to produce intermediate or by-product compounds that were affected by biotic factors (*i.e.* enzyme  secretion, toxicology metabolism, and cell growth) and abiotic factors (*i.e.* physico-chemical condition).

Generally, the pathway of xenobiotic biotransformation involves two phases. Firstly, reduction, oxidation and hydrolysis, which usually result in a small increase in hydrophilicity. Secondly, conjugation, methylation and acylation, which may or may not be preceded by the first phase. The enzymes involved in the process tend to have broad and overlapping substrate specificities [30].

## 3.6. Toxicity study

### 3.6.1. Phytotoxicity

Table 2 shows the results of the phytotoxicity test using three agricultural seeds that are common in Malaysia: *Vigna radiata*, *Arachis hypogaea* and *Vigna unguiculata*. Each specimen gave a different

**Table 2.** Phytotoxicity studies of metabolites extracted from MR biodegradation in the different culture treatments (static and shaking) on three specimens. Values are means of three replicates $\pm$ s.e.m.

| specimen | parameter studied | distilled water | control MR | static metabolite extract | shaking metabolite extract |
|---|---|---|---|---|---|
| *Vigna radiata* | germination rate (%) | 100 ± 5.77 | 100 ± 5.77 | 100 ± 5.77 | 100 ± 5.77 |
| | plumule length (cm) | 12.71 ± 1.64 | 10.67 ± 0.40 | 16.83 ± 1.11** | 16.50 ± 0.61** |
| | radicle length (cm) | 8.37 ± 1.00 | 8.41 ± 1.91 | 11.07 ± 1.04 | 11.50 ± 1.23 |
| *Arachis hypogaea* | germination rate (%) | 100 ± 5.77 | 70 ± 2.89** | 80 ± 2.89 | 100 ± 5.77 |
| | plumule length (cm) | 3.57 ± 0.12 | 2.23 ± 0.24*** | 4.37 ± 0.35* | 4.17 ± 0.25 |
| | radicle length (cm) | 3.85 ± 0.54 | 3.03 ± 0.45 | 3.54 ± 0.41 | 4.10 ± 0.36 |
| *Vigna unguiculata* | germination rate (%) | 100 ± 5.77 | 60 ± 2.89*** | 80 ± 2.89* | 80 ± 2.89* |
| | plumule length (cm) | 14.13 ± 1.00 | 9.26 ± 1.70* | 13.5 ± 1.74 | 9.50 ± 1.39* |
| | radicle length (cm) | 9.43 ± 1.23 | 6.23 ± 0.49** | 8.97 ± 0.77 | 6.37 ± 0.35** |

Significantly different from control at *$p < 0.05$, **$p < 0.01$, ***$p < 0.001$ by one-way ANOVA with Tukey–Kramer multiple comparisons test.

**Table 3.** Growth inhibition halo (millimetre) of metabolites extracted from MR biodegradation in the different culture treatments (static and shaking) and untreated MR against some bacteria. n.d., not detected. Values are means of three replicates $\pm$ s.e.m.

| | halo of inhibition zones (mm) | | | |
|---|---|---|---|---|
| | Gram-positive bacteria | | Gram-negative bacteria | |
| compound | *E. coli* | *Bacillus megaterium* | *Pseudomonas aeruginosa* | *Staphylococcus aureus* |
| tetracycline (positive control) | 19.67 ± 0.58 | 22.67 ± 0.58 | — | 33.33 ± 0.58 |
| NaCl 0.9% (negative control) | 00.00 ± 0.00 | 00.00 ± 0.00 | 00.00 ± 0.00 | 00.00 ± 0.00 |
| control MR | 20.33 ± 0.58 | 21.00 ± 1.00* | 18.33 ± 0.58*** | 21.00 ± 1.00*** |
| static metabolite extract | n.d.*** | n.d.*** | n.d. | n.d.*** |
| shaking metabolite extract | n.d.*** | n.d.*** | n.d. | n.d.*** |

Significantly different from positive control at *$p < 0.05$, **$p < 0.01$, ***$p < 0.001$ by one-way ANOVA with Tukey–Kramer multiple comparisons test.

susceptibility to the influence of daily watering using dye (untreated) and its extracted degraded metabolite (treated). According to the results, untreated MR significantly decreased the germination rate and plumule length of both of *A. hypogaea* and *V. unguiculata*, and also the radicle length of *V. unguiculata*. The tolerance level of the untreated dye was *V. radiata* > *A. hypogaea* > *V. unguiculata*. Daily watering using extracted degraded metabolite resulted in the increased plumule length of *V. radiata* ($p < 0.01$) and *A. hypogaea* ($p < 0.05$). But in *V. unguiculata* seeds, a metabolite of shaking culture inhibited the germination rate while static culture caused a significant decrease in all growth parameters. Nevertheless, the germination rate of *V. unguiculata* decreased more significantly in untreated dye than daily watering with metabolites from static culture.

This study revealed that the degraded metabolites, from either static or shaking treatment after biodegradation of the dye, were less toxic than the parent untreated dye, as indicated by the increase in the plumule and radicle length. The degraded metabolite compounds were not identified as

hazardous degradation products of azo dyes, investigated by the European Community (1994) as was described in [21].

### 3.6.2. Microbial toxicity

As shown in table 3, untreated MR showed lower significantly effects in the microbial growth in comparison with the positive control (tetracycline) for *B. megaterium* ($p < 0.05$) and *S. aureus* ($p < 0.001$), but significantly higher effects for *P. aeruginosa* ($p < 0.001$). An aliquot of untreated MR showed the zone of inhibition that indicated the presence of toxins that inhibit microbial growth, whereas the extracted metabolites of MR degradation from both static and shaking culture did not show any growth inhibition. This free halo zone on the agar plate indicated that the degraded metabolite products from static and shaking culture were not toxic to all bacterial samples. These results corroborate the resistance of *B. megaterium* against an extract of MR degraded by *Bacillus circulans* and *Providencia alcalifaciens* [31].

# 4. Conclusion

*Lysinibacillus fusiformis* strain W1B6 demonstrated high decolorization efficiency with the production of azoreductase on MR azo dye under static and shaking conditions. The main fragmented metabolites from the biodegradation of MR dye were 2-ABA and DMPD with different concentrations produced in both conditions. Methyl anthranilate (2-aminobenzoic acid, methyl ester) was also found as a degraded metabolite through the biotransformation of 2-ABA. The production of biodegradable and less toxic metabolites as a result of the action of this strain shows its potential for efficient azo dye biodegradation.

Authors' contributions. I.P.S. designed the study plan and carried out the research. K.S. provided intellectual input into the design experiments, interpretation, review and editing. Both authors drafted the manuscript and gave final approval for publication.

Data accessibility. Data are available from the Dryad Digital Repository: https://doi.org/10.5061/dryad.2cb0g0h [32].

Competing interests. We declare we have no competing interests.

Funding. This work was financially supported by an IPPP grant from the University of Malaya (project no. PG 057-2013A).

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
