## [Reviewer comments · Royal Society Open Science]

Review History

RSOS-190152.R0 (Original submission)

Review form: Reviewer 1 (Saurabh Gangola)

Is the manuscript scientifically sound in its present form?

Yes

Are the interpretations and conclusions justified by the results?

Yes

Is the language acceptable?

No

Is it clear how to access all supporting data?

Yes

Do you have any ethical concerns with this paper?

No

Have you any concerns about statistical analyses in this paper?

No

Recommendation?

Accept with minor revision (please list in comments)

Comments to the Author(s)

1-All the results are total data based. No dye decolorization photographs and Toxicity results are shown.

2- Some sentences are not clear "A 100 ppm metabolite extracted and untreated MR was used for the test." and grammatical errors are also there.

3- Results and conclusion is not too strong.

Review form: Reviewer 2

Is the manuscript scientifically sound in its present form?

Yes

Are the interpretations and conclusions justified by the results?

Yes

Is the language acceptable?

Yes

Is it clear how to access all supporting data?

Yes

Do you have any ethical concerns with this paper?

No

Have you any concerns about statistical analyses in this paper?

No

Recommendation?

Accept with minor revision (please list in comments)

Comments to the Author(s)

This manuscript reports the results on a study of methyl red dye decolorisation by *Lysinibacillus fusiformis* bacteria. The study reports interesting, though quite specific results. This manuscript can be published after minor revisions noted:

- please provide more details on how the *Lysinibacillus fusiformis* species was identified.
- add scale bars to Fig 1 and check magnification (100[is not correct)
- revise Fig.4 to make it clearly visible
- the toxicity study models (plants and bacteria), why did you use them, not human cell cultures, which would be better for this study. Please provide some rationale on this

Review form: Reviewer 3

Is the manuscript scientifically sound in its present form?

No

Are the interpretations and conclusions justified by the results?

No

Is the language acceptable?

No

Is it clear how to access all supporting data?

Yes

Do you have any ethical concerns with this paper?

No

Have you any concerns about statistical analyses in this paper?

No

Recommendation?

Reject

Comments to the Author(s)

This manuscript is poorly written to be considered suitable for publication. Many sentences have not sense, they are not written in intelligible English. It is very difficult to follow and understand the manuscript. Therefore, I do not recommend its publication.

Decision letter (RSOS-190152.R0)

24-Apr-2019

Dear Dr SIMARANI:

Title: Comparative static (anoxic) and shaking culture of metabolite derived from Methyl Red degradation by *Lysinibacillus fusiformis* strain W1B6
Manuscript ID: RSOS-190152

The editor assigned to your manuscript has now received comments from reviewers. We would like you to revise your paper in accordance with the referee and Subject Editor suggestions which can be found below (not including confidential reports to the Editor). Please note this decision does not guarantee eventual acceptance.

Please submit your revised paper before 17-May-2019. Please note that the revision deadline will expire at 00.00am on this date. If we do not hear from you within this time then it will be assumed that the paper has been withdrawn. In exceptional circumstances, extensions may be

possible if agreed with the Editorial Office in advance. We do not allow multiple rounds of revision so we urge you to make every effort to fully address all of the comments at this stage. If deemed necessary by the Editors, your manuscript will be sent back to one or more of the original reviewers for assessment. If the original reviewers are not available we may invite new reviewers.

Please also include the following statements alongside the other end statements. As we cannot publish your manuscript without these end statements included, if you feel that a given heading is not relevant to your paper, please nevertheless include the heading and explicitly state that it is not relevant to your work.

- Acknowledgements

On behalf of the Subject Editor Professor Anthony Stace and the Associate Editor Dr Andrew Harned.

RSC Associate Editor:

Comments to the Author:

The work reported in this manuscript could have some interest to our readership, but the presentation is currently below our standard. I recommend the authors revise the manuscript in order to address the concerns raised by reviewers 1 and 2. They must also completely revise the manuscript with regard to grammar, readability, and communicating their message. Seeking assistance in the latter is highly recommended.

RSC Subject Editor:
Comments to the Author:
(There are no comments.)

Reviewers' Comments to Author:
Reviewer: 1

Comments to the Author(s)

1-All the results are total data based. No dye decolorization photographs and Toxicity results are shown.

2- Some sentences are not clear"A 100 ppm metabolite extracted and untreated MR was used for the test." and grammatical errors are also there.

3- Results and conclusion is not too strong.

Reviewer: 2

Comments to the Author(s)

This manuscript reports the results on a study of methyl red dye decolorisation by *Lysinibacillus fusiformis* bacteria. The study reports interesting, though quite specific results. This manuscript can be published after minor revisions noted:

- please provide more details on how the *Lysinibacillus fusiformis* species was identified.
- add scale bars to Fig 1 and check magnification (100[is not correct)
- revise Fig.4 to make it clearly visible
- the toxicity study models (plants and bacteria), why did you use them, not human cell cultures, which would be better for this study. Please provide some rationale on this

Reviewer: 3

Comments to the Author(s)

This manuscript is poorly written to be considered suitable for publication. Many sentences have not sense, they are not written in intelligible English. It is very difficult to follow and understand the manuscript. Therefore, I do not recommend its publication.

Author's Response to Decision Letter for (RSOS-190152.R0)

See Appendix A.

RSOS-190152.R1 (Revision)

Review form: Reviewer 2

Is the manuscript scientifically sound in its present form?

Yes

Are the interpretations and conclusions justified by the results?

Yes

Is the language acceptable?

Yes

Do you have any ethical concerns with this paper?

No

Recommendation?

Accept as is

Comments to the Author(s)

The revised MS can be published as is

Decision letter (RSOS-190152.R1)

08-Jul-2019

Dear Dr SIMARANI:

Title: Comparative static (anoxic) and shaking culture of metabolite derived from Methyl Red degradation by *Lysinibacillus fusiformis* strain W1B6

Manuscript ID: RSOS-190152.R1

It is a pleasure to accept your manuscript in its current form for publication in Royal Society Open Science. The chemistry content of Royal Society Open Science is published in collaboration with the Royal Society of Chemistry.

The comments of the reviewer(s) who reviewed your manuscript are included at the end of this email. I apologise this has taken longer than usual.

Yours sincerely,

Dr Laura Smith

Publishing Editor, Journals

Royal Society of Chemistry

Thomas Graham House

Science Park, Milton Road

Cambridge, CB4 0WF

Royal Society Open Science - Chemistry Editorial Office

On behalf of the Subject Editor Professor Anthony Stace and the Associate Editor Dr Andrew Harned.

RSC Associate Editor:
Comments to the Author:
The revised manuscript can now be accepted.

RSC Subject Editor:
Comments to the Author:
(There are no comments.)

Reviewer(s)' Comments to Author:
Reviewer: 2

Comments to the Author(s)
The revised MS can be published as is

Appendix A

RSC Associate Editor:

Comments to the Author(s)

The work reported in this manuscript could have some interest to our readership, but the presentation is currently below our standard. I recommend the authors revise the manuscript in order to address the concerns raised by reviewers 1 and 2. They must also completely revise the manuscript with regard to grammar, readability, and communicating their message. Seeking assistance in the latter is highly recommended.

It has been revised

Reviewer: 1

Comments to the Author(s)

1-All the results are total data based. No dye decolorization photographs (DONE. It has done by Figure 4) and Toxicity results are shown (DONE).

The toxicity results are available in Table 2 and 3 and also in this link:

<https://datadryad.org/review?doi=doi:10.5061/dryad.2cb0g0h>

2- Some sentences are not clear "A 100 ppm metabolite extracted and untreated MR was used for the test." (DONE *It has already revised in main document and Table 1 and 2*) and grammatical errors are also there (DONE, *it has been revised*).

3- Results and conclusion is not too strong. (DONE, *it has been revised and added*).

Reviewer: 2

Comments to the Author(s)

This manuscript reports the results on a study of methyl red dye decolorisation by *Lysinibacillus fusiformis* bacteria. The study reports interesting, though quite specific results. This manuscript can be published after minor revisions noted:

- please provide more details on how the *Lysinibacillus fusiformis* species was identified. *It has already been written on sub chapter 3.2 that isolation and identification of the strain was according to Reference no 13. (DONE)*

- add scale bars to Fig 1 and check magnification (100[is not correct). *It has been revised to 1000 magnification with addition the scale bars (DONE)*

- revise Fig.4 to make it clearly visible. *DONE and revised as Figure 6*

- the toxicity study models (plants and bacteria), why did you use them, not human cell cultures (as a recommendation), which would be better for this study. Please provide some rationale on this.

This is because of the design of the study involved textile wastewater effluent where the plant might be the first contact from the discharge area of the effluent. The use of human cell cultures as toxicity model could be a recommendation for the further study.